# Kinetic Characteristics of Curcumin and Germacrone in Rat and Human Liver Microsomes: Involvement of CYP Enzymes

**DOI:** 10.3390/molecules27144482

**Published:** 2022-07-14

**Authors:** Shaofeng Su, Hongxian Wu, Jingfan Zhou, Guangwei Yuan, Haibo Wang, Jie Feng

**Affiliations:** 1School of Pharmaceutical Sciences, Guangxi Medical University, Nanning 530021, China; 18587761937@163.com (S.S.); wuhongxian0823@163.com (H.W.); zhjf@sr.gxmu.edu.cn (J.Z.); 2Guangxi-ASEAN Food Inspection and Testing Center, Nanning 530029, China; ygw@gacfdc.cn

**Keywords:** cytochrome P450 enzymes, UPLC–MS/MS, curcumin, germacrone, drug–drug interaction

## Abstract

Curcumin and germacrone, natural products present in the Zingiberaceae family of plants, have several biological properties. Among these properties, the anti-NSCLC cancer action is noteworthy. In this paper, kinetics of the two compounds in rat liver microsomes (RLMs), human liver microsomes (HLMs), and cytochrome P450 (CYP) enzymes (CYP3A4, 1A2, 2E1, and 2C19) in an NADPH-generating system in vitro were evaluated by UP-HPLC–MS/MS (ultrahigh-pressure liquid chromatography–tandem mass spectrometry). The contents of four cytochrome P450 (CYP) enzymes, adjusting by the compounds were detected using Western blotting in vitro and in vivo. The t_1/2_ of curcumin was 22.35 min in RLMs and 173.28 min in HLMs, while 18.02 and 16.37 min were gained for germacrone. The V_max_ of curcumin in RLMs was about 4-fold in HLMs, meanwhile, the V_max_ of germacrone in RLMs was similar to that of HLMs. The single enzyme t1/2 of curcumin was 38.51 min in CYP3A4, 301.4 min in 1A2, 69.31 min in 2E1, 63.01 min in 2C19; besides, as to the same enzymes, t1/2 of germacrone was 36.48 min, 86.64 min, 69.31 min, and 57.76 min. The dynamic curves were obtained by reasonable experimental design and the metabolism of curcumin and germacrone were selected in RLMs/HLMs. The selectivities in the two liver microsomes differed in degradation performance. These results meant that we should pay more attention to drugs in clinical medication–drug and drug–enzyme interactions.

## 1. Introduction

As we all know, the nonsmall cell lung cancer (NSCLC) is one of the severest malignancies dangerous for human’s health, whose incidence and mortality are both increasing rapidly in recent years. It severely endangers human health and life [1]. Curcumin and germacrone (Figure 1) are the main active ingredients in the Zingiberaceae family of plants such as *Curcuma aromatica* Salisb., *C. longa* L. [2]. It was reported that curcumin and germacrone showed strong anti-NSCLC cancer effects and are easy to be obtained from natural medicine [3,4,5,6]. NSCLC cancer requires an expensive clinical treatment [7], so curcumin and germacrone are promising for the development of new NSCLC drugs. Preclinical studies are always useful for a prospective new drug [8,9]; meanwhile, the kinetics offer use as a model in preclinical research [10,11]. The blood pharmacokinetics and urodynamics with curcumin or germacrone in rats were reported [12,13,14], but no more studies on kinetics of curcumin and germacrone with the liver microsomes metabolism were conducted.

The liver microsomes are rich in CYP, associating with drug metabolism and biotransformation [15,16]. The candidate drugs pharmacokinetic properties and characteristics in vivo can be predicted through detecting the metabolic rates and enzyme kinetics characteristics in vitro [17,18,19]. It is of great significance to clarify the metabolic enzyme subtypes of drugs in liver microsomal metabolism in vitro, and the mechanisms of drugs combine to achieve synergistic effects and detoxification [20,21,22]. According to the literatures, curcumin has been evaluated the metabolites with an ^18^O isotope labeling strategy in HLMs [23]. The interaction of zederone and germacrone with the main human liver CYPs, the sensor regulators, and the potential protected effect for liver were explored [24]. It is well known that the metabolism of drug candidates in humans may different from that of other species; meanwhile, the variety of metabolites and the concentrations can be potentially affected by the bioactivities and toxicities in vivo. Compared with in vivo, the interference of endogenous substances can be avoided in vitro and can also be observed the interaction between agent and target. Hence, in this paper, the kinetics of curcumin and germacrone were investigated in RLMs and HLMs in vitro using UPLC–MS/MS.

CYP enzymes can play an antiapoptotic role on bovine aortic endothelial cells through MAPK (ERK1/2) and PI3K/Akt pathway [25]. It is of great significance to explore the relationships between curcumin or germacrone with enzymes for the treatment of NSCLC in vitro and in vivo. Therefore, the important component of CYP enzymes of CYP3A4, 1A2, 2E1, and 2C19 were selected to incubate with curcumin or germacrone, and protein expression was analyzed using Western blotting in vitro and in vivo.

Compared with in vivo, in vitro studies require fewer animals and avoid interference from endogenous substances [26]. Currently, the models for studying drugs metabolism in vitro mainly include subcell line models (such as microsomes, cytosol, and S9), recombinant enzyme line models, tissue section models, and in vitro animal models [27]. CYP, relating to biotransformation, has a large proportion in the liver microsomes. According to the statistics, about 90% of drugs biotransformation need to go through the CYP system, meaning that CYP enzyme is the main prerequisite for the biological transformation, and difference enzymes are controlled by genetic diversity. The metabolic pathways and metabolites of drugs may be different in vitro and in vivo, and polymorphisms are an important characteristic of CYP enzymes, leading to the individual differences in drug reactions [28,29]. The content and activity of CYP enzymes can be affected by different agent, which can affect the metabolism, continuously cause metabolic drug interactions.

As a potential agent for NSCLC treatment, there are need to more studies on kinetic degradation in human single recombinant CYP enzymes of CYP3A4, 1A2, 2E1, 2C19, RLMs, and HLMs and enzyme kinetic selectivity in RLMs and HLMs. The molecular mechanisms of CYP enzymes activity and the expression levels by curcumin or germacrone were detected in vitro and in vivo. It is possible to develop new target drugs and guide them to preclinical treatment.

## 2. Results

### 2.1. Assay Validation

The results shown that their RSD < 5%, and the stock solutions curcumin or germacrone were stabled in 1 month. The retention times of curcumin or germacrone was 12.24 or 14.86 min, the topical chromatogram were shown in (Figure 2).

The linear regression results of matrix-matched and standard solution calibration curves (0.01–120.00 μM) of curcumin and germacrone showed good linear relationship, the correlation coefficient (*r*^2^) were 0.9948–0.9998. The LOD and LOQ of curcumin were 0.018 and 0.150 μM in RLMs, 0.025 and 0.130 μM in HLMs, meanwhile germacrone were 0.032 and 0.090 μM in RLMs, 0.026 and 0.120 μM in HLMs, respectively, see Appendix A. The accuracy of curcumin and germacrone in RLMs were 85.53–98.26%, the precision were 1.22–3.04% (RSD) at three concentrations level. The accuracy of curcumin and germacrone in HLMs were 88.42–96.47%, the precision were 1.08–2.62% (RSD) at three concentrations level, see Appendix A. The recovery of curcumin and germacrone in RLMs were 95.36–97.36% and 98.63–99.96%, the relation RSD were 2.12–3.42 and 1.56–2.67; the recovery of curcumin and germacrone in HLMs were 97.12–100.46% and 98.68–102.32%, the relation RSD were 1.92–2.72 and 2.34–3.42, respectively, see in Appendix A.

### 2.2. Degradation of Curcumin and Germacrone in RLMs and HLMs

The curcumin or germacrone concentrations and incubation times were showed in (Figure 3). Curcumin and germacrone had almost no metabolic degradation in RLMs and HLMs without NADPH. The initial concentrations for the degradation incubation for 120 min in RLMs or HLMs with NADPH were 50 μM curcumin and 60 μM germacrone, respectively. The speed constant K was reckoned by the equation C = C_0_e^−kt^ fitted by regression analysis. The depletion half-life t_1/2_ was defined as t_1/2_ = ln 2/k. C_0_ was defined as the initial concentration (μM), and its concentration (μM) varying with t (min) was defined as C. The regression equation in the RLMs in vitro of curcumin was C = (49.76 ± 0.11) e^−(0^^.031 ± 0^^.0014)t^ (*r*^2^ = 0.9935), and germacrone was C = (58.57 ± 0.72) e^−(0^^.038 ± 0^^.0011)t^ (*r*^2^ = 0.9945); the t_1/2_ of curcumin and germacrone were 22.35 min and 18.02 min. Meanwhile, the regression equation in HLMs in vitro of curcumin was C = (52.51 ± 0.09) e^−(0^^.004 ± 0^^.0056)t^ (*r*^2^ = 0.9633), and germacrone was C = (60.78 ± 0.13) e^−(0^^.042 ± 0^^.0026)t^ (*r*^2^ = 0.9842); the t_1/2_ of curcumin and germacrone were 173.28 min and 16.37 min, respectively.

### 2.3. Enzyme Kinetics of Curcumin and Germacrone

The curcumin or germacrone was incubated for 15 min in RLMs or HLMs, and the K_m_ and V_max_ were determined. The Michaelis-Menten equation curves with the curcumin (1, 5, 10, 30, 50, 80 and 100 μM) or germacrone concentrations (1, 5, 10, 30, 60, 80 and 100 μM) were shown in (Figure 4). The Lineweaver-Burk equation, 1/V = (K_m_/V_max_) (1/S) + 1/V_max_, a straight line of a slope of K_m_/V_max_ was obtained by double reciprocal plot of 1/V and 1/S. The degradation of curcumin and germacrone were conformed to the first-order kinetics reaction in RLMs and HLMs. The kinetic parameters of curcumin and germacrone degradation in RLMs and HLMs were shown in the (Table 1).

### 2.4. Degradation of Curcumin and Germacrone in CYP3A4, 1A2, 2E1 and 2C19

The four CYP enzymes of CYP3A4, 1A2, 2E1 and 2C19 were reacted with 5 μM curcumin or germacrone to explore the degradation between the compounds with a single enzyme in vitro.

The results were showed in (Figure 5). The regression equation in the CYP3A4 of curcumin was C = (5.08 ± 0.58) e^−(0.018 ± 0.0035)t^ (*r*^2^ = 0.9967), and germacrone was C = (4.91 ± 0.085) e^−(0.019 ± 0.0064)t^ (*r*^2^ = 0.9970). The t_1/2_ of curcumin and germacrone were 38.51 min and 36.48 min, respectively.

The regression equation in CYP1A2 of curcumin and germacrone were C = (5.17 ± 0.04) e^−(0.0023 ± 0.0006)t^ (*r*^2^ = 0.9153), C = (4.75 ± 0.087) e^−(0.008 ± 0.001)t^ (*r*^2^ = 0.9922), the t_1/2_ were 301.4 min and 86.64 min, respectively.

The regression equations in CYP2E1 of curcumin and germacrone were C = (4.78 ± 0.041) e^−(0.01 ± 0.0024)t^ (*r*^2^ = 0.9894), C = (4.74 ± 0.010) e^−(0.01 ± 0.0017)t^ (*r*^2^ = 0.9939), the t_1/2_ were 69.31 min and 69.31 min, respectively.

The regression equations in CYP2C19 of curcumin and germacrone were C = (4.97 ± 0.407) e^−(0.011 ± 0.0017)t^ (*r*^2^ = 0.9972), C = (4.96 ± 0.067) e^−(0.012 ± 0.0032)t^ (*r*^2^ = 0.9970), the t_1/2_ were 63.01 min and 57.76 min, respectively.

### 2.5. Effects of Curcumin or Germacrone on CYP3A4, 1A2, 2E1 and 2C19 Activity

The effect of curcumin or germacrone on CYP enzymes activity in vitro were showed in (Figure 6). According to the data, when the concentrations of curcumin or germacrone (0, 5, 10, 30, 50 and 80 μM) were used, the IC_50_ value for inhibition activity in corresponding enzymes of CYP3A4, 1A2, 2E1 and 2C19 were 24.27 μM, 92.60 μM, 172.05 μM and 11.86 μM for curcumin, and 3.57 μM, 78.06 μM, 120.35 μM and 17.84 μM for germacrone, respectively.

### 2.6. Pharmacokinetic Study with Probe Substrate In Vivo

The results showed (Figure 7) that with the increase concentrations of curcumin or germacrone after i.g. administrated, midazolam, phenacetin, chlorzoxazone and omeprazole expressed different metabolic rates. When the curcumin concentrations increased, the metabolic rates of these exclusive substrates decreased, meaning that curcumin shown an inhibited effect on corresponding enzymes activity. Conversely, germacrone did the opposition. As the concentrations of germacrone rised, the metabolic rates of the exclusive substrates increased, meaning that germacrone enhanced the enzymes activity.

### 2.7. The Protein Expressions Levels of Curcumin and Germacrone on CYP Enzymes In Vitro

As shown in (Figure 8), the protein levels of CYP3A4, 1A2, 2E1 and 2C19 were significantly decreased in curcumin groups, compared to control group (*p* < 0.05), and the protein expression levels were decreased. Germacrone, however, did the opposite. As the concentrations of germacrone increased, the levels of CYP3A4, 1A2, 2E1 and 2C19 were significantly increased. Both two compounds protein levels shown a dose-dependent manner.

### 2.8. Effect of Curcumin and Germacrone on the Protein Levels of CYP Enzymes In Vivo

After 7-days curcumin treatment, compared with control group, the CYP3A4, 1A2, 2E1 and 2C19 protein levels were decreased in low-, medium- and high-dose group rats. As to the germacrone treatment, compared with control group, the CYP3A4 1A2, 2E1 and 2C19 protein levels were increased in low-, medium- and high-dose group rats. The protein expressions in vitro and in vivo had the same tendency for the two compounds. The results showed in Figure 9.

## 3. Discussion

Curcumin and germacrone are thought to possess a lot of beneficial bioactivities, including anti-NSCLC cancer and antiviral effects, and obtained from raw materials [30,31,32,33]. So, curcumin and germacrone are very promising to develop the new NSCLC drugs. Enriching the species difference information of selective of curcumin or germacrone between human and experimental animals are of great significance in new drug development. Hence, the kinetic degradation in human single recombinant CYP enzymes of CYP3A4, 1A2, 2E1, 2C19, enzyme kinetics of curcumin or germacrone with RLMs or HLMs were examined in present study. There are reports that the relationship between CYP enzymes with drugs can be provided a better guidance for clinical drug combination [34,35]. Therefore, we went on exploring the potential influence of two compounds on the CYP3A4, 1A2, 2E1 and 2C19 enzyme in order to provide more insight on drug-drug combination in clinic.

The t_1/2_ as a prerequisite for medicinal properties, and it was need to be more than 30 min in the liver microsomes [36,37]. To obtain the kinetic degradation between curcumin and germacrone with liver microsomes, curcumin and germacrone incubation with NADPH regeneration system in RLMs or HLMs for 120 min. The t_1/2_ of curcumin or germacrone with RLMs or HLMs were first investigated in this paper. The results showed that the degradation of curcumin or germacrone were related to substrate concentrations and incubation time. There was little metabolic degradation of curcumin or germacrone in liver microsomes without NADPH regeneration system, meaning that, the degradation of curcumin and germacrone were NADPH-dependent. The degradation curve of curcumin or germacrone incubation in RLMs and HLMs with NADPH indicated that the concentrations of curcumin or germacrone were different at the same time point. The t_1/2_ of curcumin and germacrone in RLMs was 22.35 and 18.02 min, while, it was 173.28 and 16.37 min in HLMs, respectively. The t_1/2_ of curcumin in HLMs was almost 8-fold than that in RLMs, meaning that curcumin affinity with RLMs stronger than that of HLMs. But the t_1/2_ of germacrone in RLMs was similar as in HLMs. From the data, the degradation of curcumin was much faster in RLMs than that in HLMs, but germacrone was stronger in HLMs than that in RLMs. The selective of curcumin markedly differed among species in RLMs and HLMs, but germacrone had a weak selective.

Kinetic analyses of curcumin or germacrone selectived in RLMs and HLMs were subsequently performed at a broad range of substrate concentrations. Both the kinetics of curcumin and germacrone in those liver microsomes fitted the Michaelis-Menten model. The V_max_ and CL_int_ for curcumin in RLMs were markedly higher than in HLMs. Inversely, the V_max_ and C_Lint_ for germacrone in HLMs were higher than in RLMs. The K_m_ for curcumin in RLMs also were markedly higher than in HLMs, but germacrone in HLMs higher than in RLMs. Contrasted with HLMs, the degradation and enzyme kinetics of curcumin had a stronger potency to deplete in RLMs, but germacrone did the opposition. There may be some possible factors that led to the results. The biological competitive inhibition, transformation, interaction between compounds led to unconformity of the affinity, metabolic capability, the chemical structures, and the selection of different degradation enzymes were all can change in RLMs and HLMs [38,39,40].

The metabolic rate of drugs are mainly determined by the enzyme activity and content. Currently reported about curcumin and germacrone on CYP enzyme activity in vitro, but no enzyme content. In inhibitory experiment with enzyme activity, curcumin inhibited effects on CYP3A4, 1A2, 2E1 and 2C19, with IC_50_ of 24.27 μM, 92.60 μM, 172.05 μM and 11.86 μM, and germacrone inhibited effects on the same CYP enzymes, with IC_50_ of 3.57 μM, 78.06 μM, 120.35 μM and 17.84 μM, respectively. The results were generally consistent with those reports, which CYP3A4, 1A2 were effected by germacrone [24,41]. This study for curcumin and germacrone on CYP enzyme can enrich their activity and content. In the metabolism with single enzyme experiment, t_1/2_ of curcumin in CYP3A4, 1A2, 2E1 and 2C19 were 38.51, 301.4, 69.31 and 63.01 min, meanwhile, t_1/2_ of germacrone in the same enzymes were 36.48, 86.64, 69.31 and 57.76 min, respectively. These results meant that CYP3A4 and 2C19 could be inhibited by curcumin or germacrone significantly, and curcumin or germacrone also was metabolized simultaneously by CYP3A4 and 2C19. Combined with the kinetics, it was reasonable assumed that the metabolic rate was affected by enzyme content. To demonstrate this assumption, the enzyme expression was analyzed using Western blotting in vitro. Further demonstrates for the metabolic rate was associated with enzyme content levels in RLMs and HLMs. Moreover, the protein expressions of CYP3A4, 1A2, 2E1 and 2C19 were decreased by curcumin, while germacrone did the opposition in vivo. The t_1/2_ of the four proprietary enzyme substrates were enhanced in curcumin but shortened in germacrone in vivo. Thus, the metabolic rate was related to content enzyme.

Drug combination always uses to improve the effective treatment and reduce toxicity in clinic, such as nintedanib, icotinib, erlotinib and pembrolizumab are frequently used as a combination drug for NSCLC patients [42,43,44,45]. Erlotinib is metabolized mainly by CYP3A4 [46]. Interestingly, curcumin had inhibitory effect on CYP3A4 enzymes. Currently, curcumin combined with afatinib in NSCLC treatment has been patented and applied in clinic. Can t_1/2_ of the erlotinib be appropriately prolonged when combined with curcumin? This study provides new insights and ideas for developing target new drugs, the combination with other cancer drugs in clinical practice.

## 4. Materials and Methods

### 4.1. Materials

The rhizome of *Curcuma aromatica* Salisb. was collected from Xinfu Hengxian in November 2016, Guangxi, China. The species was identified by Jie Feng (one of the authors) and Maoxiang Lai (Guangxi Academy of Chinese Medicine and Pharmaceutical Science, Nanning, China), and the voucher specimens (No. 20161106) were deposited at the School of Pharmaceutical Sciences, Guangxi Medical University, Nanning, China. Phenobarbital sodium was bought from Shanghai SPH New Asia Pharmaceutical Co. Ltd. (PS, Shang Hai, China, H31020501). Nicotinamide adenine dinucleotide phosphate disodium salt (NADPNa2, 116C023), Glibenclamide (HW20G1402-2, IS), nicotinamide adenine dinucleotide phosphate (NADH, 811B039), Glucose-6-phosphate (D8090), and Glucose-6-phosphate dehydrogenase (1000 IU/1.36 mg, G8020) were purchased from Solarbio (Beijing, China). Midazolam, testosterone, omeprazole, phenacetin, and chlorzoxazone were purchased from Sigma Company (Chengdu maide biotechnology co. LTD, Chengdu, China). Human liver microsomes (HLMs) and human single recombinant CYP enzymes (CYP3A4, 1A2, 2E1, 2C19) were purchased from RILD (JPXY: Research Institute for Liver Diseases Co., Ltd., Shanghai, China). Ultrapure water was prepared with a Milli-Q water purification system (Billerica, MA, USA). The other analytical reagents were obtained from commercial sources.

### 4.2. Extraction, Isolation and Identification

Shade-dried, powdered rhizomes of *C. aromatica* was extracted with 10 times 70% hydroalcoholic ethanol by heating reflux for three times to obtain the filtrate, then was concentrated and suspended in H_2_O. Sequentially partitioned with 5 times petroleum ether (PE), EtOAc. The organic solvents were removed under vacuum to give PE and EtOAc extracts, respectively. The EtOAc extract was separated on a silica gel column using CHCl_3_/MeOH (99:1 to 0:100) to yield six fractions (E1–E6). Fr E4–E6 were respectively repeated subjected to ODS column, Sephadex LH-20, semipreparative HPLC, and PTLC (preparative thin layer chromatography) to afford compounds curcumin and germacrone. Their structures were identified by comparison with the standard substance, which were bought from the National Instituted for the Control of Pharmaceutical and Biological Products of China (Beijing, China. Curcumin: PCS0359; Germacrone: PCS0537), curcumin and germacrone were confirmed by HPLC with purity >98.0%.

### 4.3. UPLC–MS/MS Condition

An Ultimate 3000 system (Thermo Fisher Scientific, Waltham, USA), which comprised a liquid pump (HPG-3400 SD), a column oven (TCC-3000 SD), an automatic sampler (WPS-3000SL), and an orbitrap detector (orbitrap Fusion Lumos), was used. The analyses were carried out on an ACUPLCBEH C_18_ column (50 mm × 2.1 mm, i.d.: 1.7 μm) acquired from Dionex at 30 °C and the temperature of automatic injector was at 4 °C. The mobile phase was consisted of 0.1% formic acid aqueous solution (solvent A) and acetonitrile (solvent B). The elution was programmed as follows: 95% solvent A for 0–2 min, 95% to 0% solvent A for 2–22 min, and 95% solvent A for 22.1–26 min. The flow rate was set at 0.3 mL/min.

A Thermo Q Exactuive^TM^ four stage rod-orbit trap LC–MS/MS system (Thermo Fisher Scientific, Waltham, USA) was used to investigate kinetics. The analyses were carried out using electrospray ionization (ESI) settings in the positive/negative ion-switching mode. The MS conditions were: ESI temperature 300 °C. Ion source voltage, 3200 V in positive ion mode and −3200 V in negative ion mode; capillary temperature was 320 °C; curtain (high-purity nitrogen) gas and sheath gas (high-purity air) pressures, 30 and 10 psi, respectively. The scanning mode is Full MS and Full MS/MS, with the quality range of 100–1500 *m*/*z*. The resolution of the first-level scan and the second-level scan were 70,000 and 17,500, respectively.

### 4.4. Method Validation

The stock solution of curcumin or germacrone stability test was performed. The standard solutions (approximately 10 mM) were stored at −20 °C and were observed at 7 and 14 days and 1 month, which were determined in parallel.

The validation procedure was carried out through the following parameters: linearity, matrix effect, accuracy and precision, limit of quantification (LOQ), and limit of detection (LOD), and each concentration was repeated three times (*n* = 3). The linearity was assessed on the curcumin or germacrone standard using the matrixes of solvent acetonitrile and blank RLMs or blank HLMs. The least-squares linear regression analysis was used to build the curcumin or germacrone calibration curves (0.05, 5, 10, 30, 80, 100, 120 μM) of seven points. The regression analysis was carried out under the analyte concentrations of the quantification ion peak areas.

The accuracy was assessed on curcumin or germacrone comparing the actual concentration to the calculated concentrate from a matrix-matched calibration curve. The reproducibility was generally expressed by the recovery rate. Three curcumin or germacrone concentration levels of 3, 10, and 80 μM with three spiked samples were evaluated for the intraday accuracy. The same concentration levels with three spiked samples on three different days were detected for the inter-day accuracy. RSD (relative standard deviation) was used to evaluate the precisions of intraday and interday values. LOD was defined as 3 times the signal-to-noise (S/N) ratio. LOQ was S/N = 10. Both LOD and LOQ used matrix-matched standards.

### 4.5. Preparation of RLMs

8 weeks SPF grade male Sprague-Dawley rats (SD, 220–250 g, certificate Number SCXK-(Gui) 2019–0002) were supplied by the animal experiment center of Guangxi Medical University (Nanning, China). This study was performed following the Chinese legislation for the use and care of experimental animals. The rats were kept in the controlled-environment breeding room (temperature 22 °C ± 2 °C, relative humidity 60 ± 5%, 12:12 h light-dark cycle). Standard laboratory chow and water were available at all times. After accommodation for one week, the rats were peritoneal injected with phenobarbital sodium saline solution at a dose of 80 mg/kg for successive four days to induce cytochrome P450 enzymes (CYPs). The rats were killed on the fifth day after fasting food for 14 h; then, the livers were quickly excised and homogenized in an ice bath. Liver obtained from 5 rats were weighed and homogenized with cold 0.01 mM PBS containing 0.25 mM sucrose; the homogenates were centrifuged at 11,600× *g* for 15 min. The supernatants were then transferred to a new tube and centrifuged at 11,600× *g* for 15 min again. The supernatants were then ultracentrifuged at 100,000× *g* for 1 h to get sediment rat liver microsomes (RLMs) protein. All the centrifuged were gone at low temperature. The concentration of RLMs was 17.23 mg/mL determined by the bicinchoninic acid (BCA) method. The microsomes were stored at −80 °C prior to use.

### 4.6. Microsomal Incubation Conditions

1 mL incubation system contained liver microsomal proteins (0.25 mg/mL, PH 7.4) or 4 pM CYP enzymes (CYP3A4, 1A2, 2E1, 2C19, respectively), 100 mM potassium phosphate buffer (pH 7.4), and NADPH solution (10 mM G-6-P, 1 IU/mL G-6-PDH, 1 mM NADP, 0.5 mM NADH, and 4 mM MgCl_2_). Curcumin or germacrone dissolved in methanol was added into the incubation system, respectively, and the methanol concentration in the final system was less than 1.0% (*v*/*v*). The reaction was initiated by adding NADPH regeneration system after pre-incubation for 10 min at 37 °C. After incubating for 120 min, the reaction was terminated by adding 3 mL ice-cold MeCN spiked with glibenclamide as IS at 100 μM, then the mixture was vortexed and centrifuged at 10,000× *g* for 10 min. The collected supernatant of the incubated system was dried by nitrogen blow and the residue was redissolved in 1 mL methanol, then was filtered by 0.22 μm membrane filter prior to LC–MS/MS analysis, and the data acquisition and quantification were conducted using Xcalibur 2.2.0 (Xcalibur software, Thermo Scientific, Waltham, MA, USA).

### 4.7. Enzyme Kinetic Assays

Different concentrations of curcumin (1, 5, 10, 30, 50, 80, and 100 μM) and germacrone (1, 5, 10, 30, 60, 80, and 100 μM) were added into the incubation system with liver microsomes (0.25 mg of protein/mL) and incubated for 15 min to evaluate the kinetic parameters. K_m_ is defined as the characteristic constant of the enzyme, which represent the half of substrate concentration at the maximum metabolic reaction velocity (V_max_). The values of V_max_ and K_m_ are based on the nonlinear regression analysis of the Michaelis-Menten equation, V = V_max_ × S/(K_m_ + S). The kinetic parameters were adopted through the following parameters: S, V_max_, and V. All of these parameters represented the substrate concentration, the maximum velocity of metabolism, and the velocity of metabolism, respectively. CL_int_ (the different ability of the organism in eliminating the particular chemical substances) is defined as V_max_/K_m_, which reflected the different ability of the organism in eliminating the particular chemical substances.

### 4.8. The Activities of Curcumin and Germacrone on CYP3A4, 1A2, 2E1 and 2C19 In Vitro

The effect of curcumin and germacrone on CYP3A4, 1A2, 2E1, and 2C19 were investigated individually. The mixed probe substrate consisted of testosterone 0.01 mM for CYP3A4, phenacetin 0.02 mM for CYP1A2, chlorzoxazone 0.02 mM for CYP2E1, and omeprazole 0.01 mM for CYP2C19, respectively. In addition, the final concentrations of curcumin or germacrone ranged from 0 to 80 μM (0, 5, 10, 30, 50, and 80). Other procedure was accorded to above step of 2.6 “Microsomal incubation conditions”.

### 4.9. Pharmacokinetic Study of the Probe Substrate In Vivo

After a week of acclimation, 42 male SD rats (200 ± 20 g) were randomly divided into four groups (*n* = 6/group): a control group (C; 0.5% CMC-Na), low dose groups (L), medium dose groups (M), and high-dose groups (H). Suspensions of curcumin or germacrone prepared with 0.5% CMC-Na solution were intragastrical (i.g.) administration to the rats. Curcumin groups were received 10, 40 and 80 mg/kg/d, and germacrone groups were received 1, 4 and 8 mg/kg/d, respectively. For consecutive seven days by i.g. administration; on the 8th day, all rats were i.g. administered with the cocktail substrates solution containing midazolam (10 mg/kg), phenacetin (10 mg/kg), chlorzoxazone (10 mg/kg), omeprazole (5 mg/kg). Blood samples (0.25 mL) were collected via eye ground puncture into heparin-containing tubes at the time intervals of 0, 5, 10, 30, 60, 90, 120, 180, 240 and 360 min. The blood samples at each time point were collected from six rats, then immediately centrifuged at 1200× *g* for 10 min at 4 °C. The plasma obtained (50 μL) was stored at −20 °C until analysis.

### 4.10. Plasma Sample Preparation

Glibenclamide, the internal standard (IS) was dissolved in methanol to prepare the approximately 1 mg/mL stock solution, the 5 μL stock solution was placed in a 10 mL EP tube as a working solution (50 ng/mL).

In a 1.5 mL polyethylene tube, a 100 μL aliquot plasma sample spiked with 100 μL of IS working solution (50 ng/mL) and 300 μL methanol, then vortexed for 3.0 min. After being centrifuged at 13,800× *g* for 10 min at 4 °C, 5 μL of the supernatant of each sample was injected into the UPLC–MS/MS system for analysis the concentrations of probe drugs.

### 4.11. The Contents of Curcumin and Germacrone on CYP3A4, 1A2, 2E1 and 2C19 with RLMs In Vitro

1 mL incubation system contained RLMs proteins (0.25 mg/mL, PH 7.4), 100 mM potassium phosphate buffer (pH 7.4), and NADPH solution (10 mM G-6-P, 1 IU/mL G-6-PDH, 1 mM NADP, 0.5 mM NADH, and 4 mM MgCl_2_). Curcumin (5, 30, 50 μΜ) and germacrone (5, 30, 60 μΜ) dissolved in methanol were added into the incubation system, incubation steps according to above 2.6 “Microsomal incubation conditions”, Then, the sediment was analyzed for the content of protein using Western blotting analysis.

### 4.12. Western Blotting Analysis

To investigate the molecular mechanisms in the activity of the CYP enzymes, the level of CYP protein in the liver of rats was examined.

In this study, the grouping and administration methods were as described in the step of “Pharmacokinetic study of the probe substrate in vivo”. On the 8th day, the blood was allowed to drain through the aorta and the liver was excised quickly, then the rats were anesthetized. The liver tissue was stored at −80 °C until analysis.

Then, a piece of the liver tissue was separately homogenized and lysed in RIPA buffer with 1% proteinase inhibitor phenylmethylsulfonyl fluoride (PMSF). After centrifugation at 13,800× *g* for 10 min at 4 °C, the supernatant of the lysates was obtained. The protein levels were quantified by a BCA Assay Kit. The samples were boiled at 100 °C for 10 min. Equal amounts of protein lysates were separated on 10% SDS-polyacrylamide gels by electrophoresis and then transferred onto nitrocellulose membranes. The membranes were blocked in 5% non-fat milk. The blots were probed using CYP3A4 antibody (1:5000, 18227-1-AP, Proteintech, Tokyo, Japan), 1A2 antibody (1:1000, 19936-1-AP, Proteintech), 2E1 antibody (1:5000, 19937-1-AP, Proteintech), and 2C19 antibody (1:1000, ab137015, Abcam), followed by incubation with the respective secondary antibodies. Bands of CYP3A4, 1A2, 2E1, 2C19, and β-actin could be detected at approximately 52, 58, 53, 56, and 43 kDa, respectively. In addition, images were captured using an Odyssey Infra-red Imaging System v3.0.16 (LI-COR Biosciences, Lincoln, NE, USA), and quantified by densitometry scanning using ImageJ Analysis software (National Institutes of Health, Bethesda, MD, USA).

### 4.13. Statistical Analysis

The experimental data was statistically processed using SPSS 19.0 software (SPSS Inc., Chicago, IL, USA). And the enzymatic kinetics profile was obtained through GraphPad Prism 5 (GraphPad Software Inc., San Diego, CA, USA). The values were represented as mean ± SD after testing the homogeneity of variance; the significance of the data was analyzed using a one-way analysis of variance (ANOVA). The differences between the groups were considered statistically significant at *p* < 0.05.

## 5. Conclusions

In summary, the degradations of curcumin in RLMs and HLMs were selective and significantly different, the degradation velocity of curcumin in HLMs was slower than that of RLMs. However, germacrone had a weak selectivity. In vivo and in vitro, curcumin could inhibit the activity and decrease the content of CYP3A4, 1A2, 2E1, and 2C19; meanwhile, germacrone could inhibit the activity of CYP3A4, 1A2, 2E1, and 2C19, but increase the content of CYP3A4, 1A2, 2E1, and 2C19. The results significantly demonstrated that both of two compounds may be developed to CYP3A4 or 2C19 target new drugs, and combination with other cancer drugs in clinical practice.

In order to further clarify the mechanism of enzymes with metabolites of curcumin or germacrone in liver microsomes to provide new insights and ideas for the NSCLC patients. The metabolites, depth mechanism, effects of germacrone or curcumin combined with lung cancer drugs are our next focus research.

## Figures and Tables

**Figure 1 molecules-27-04482-f001:**
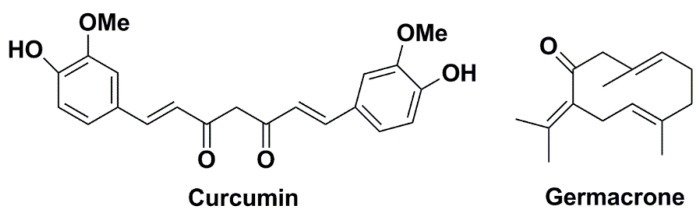
Chemical structures of curcumin and germacrone.

**Figure 2 molecules-27-04482-f002:**
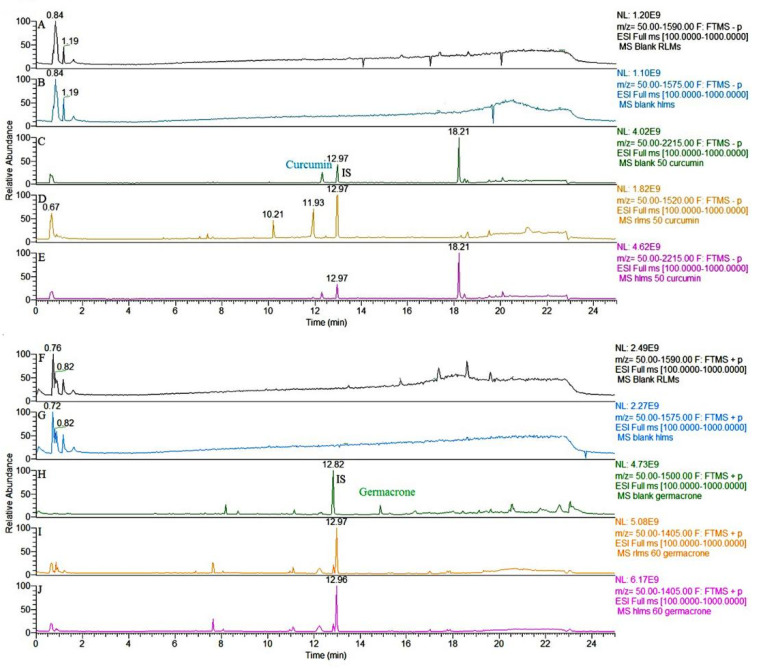
Representative HPLC–MS/MS chromatograms. (**A**,**F**): blank RLMs; (**B**,**G**): blank HLMs; (**C**,**H**): standard curcumin and germacrone (50 and 60 μM); (**D**,**I**): extract from RLMs with 50 μM curcumin and 60 μM germacrone after incubation for 120 min with NADPH; (**E**,**J**): extract from HLMs with 50 μM curcumin and 60 μM germacrone after incubation for 120 min with NADPH.

**Figure 3 molecules-27-04482-f003:**
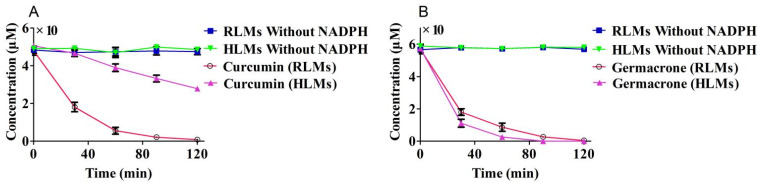
Degradation of curcumin (**A**), 50 μM and germacrone (**B**), 60 μM in RLMs and HLMs (*n* = 3).

**Figure 4 molecules-27-04482-f004:**
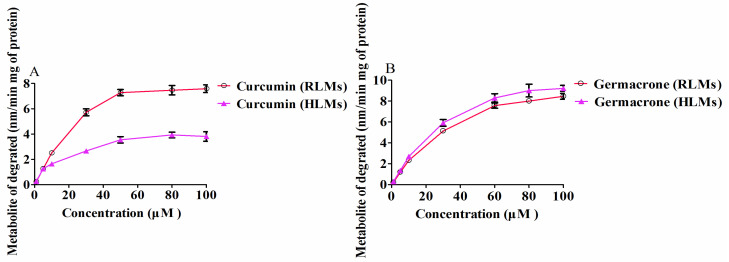
Enzyme kinetics of curcumin (**A**) and germacrone (**B**) in RLMs and HLMs (*n* = 3).

**Figure 5 molecules-27-04482-f005:**
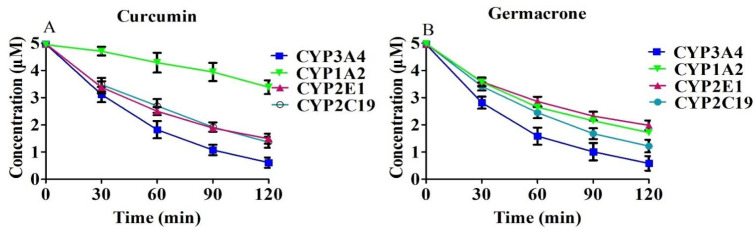
Degradation of curcumin (**A**), 5 μM and germacrone (**B**), 5 μM in CYP3A4, 1A2, 2E1 and 2C19 in vitro (*n* = 3).

**Figure 6 molecules-27-04482-f006:**
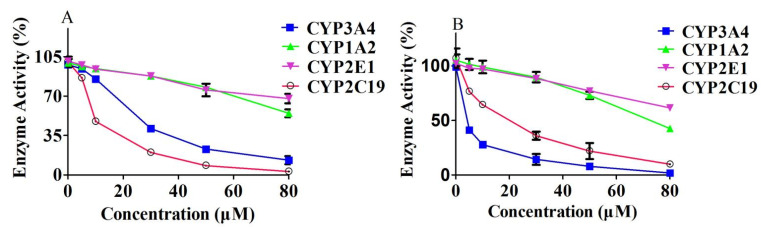
The effects of curcumin (**A**) or germacrone (**B**) on CYP3A4, 1A2, 2E1, and 2C19 activity in vitro (*n* = 3).

**Figure 7 molecules-27-04482-f007:**
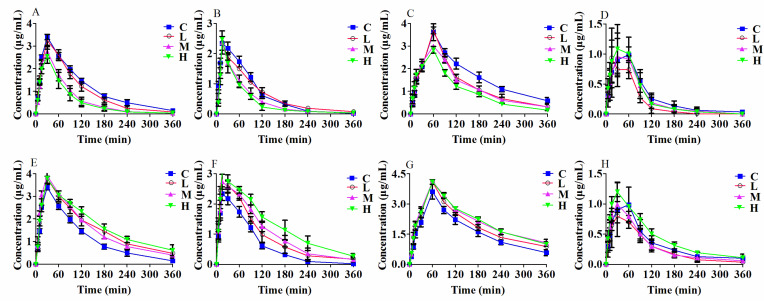
Plasma concentration-time curves of midazolam (**A**), phenacetin (**B**), chlorzoxazone (**C**), omeprazole (**D**) after i.g. administrated with curcumin (C: control, L: 10, M: 40, H: 80 mg/kg/d), and midazolam (**E**), phenacetin (**F**), chlorzoxazone (**G**), omeprazole (**H**) after i.g. administrated with germacrone (C: control, L: 1, M: 4, H: 8 mg/kg/d) in rats (mean ± SD, *n* = 6).

**Figure 8 molecules-27-04482-f008:**
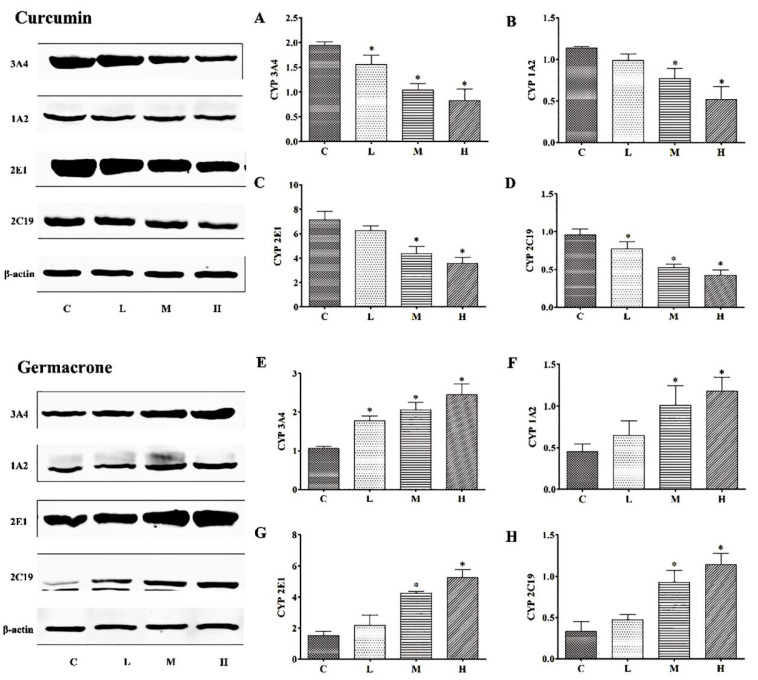
The expression levels of CYP3A4 (**A**), 1A2 (**B**), 2E1 (**C**) and 2C19 (**D**) were decreased in curcumin groups and CYP3A4 (**E**), 1A2 (**F**), 2E1 (**G**) and 2C19 (**H**) were increased in germacrone groups in vitro (mean ± SD, *n* = 6, * *p* < 0.05 vs. control rats).

**Figure 9 molecules-27-04482-f009:**
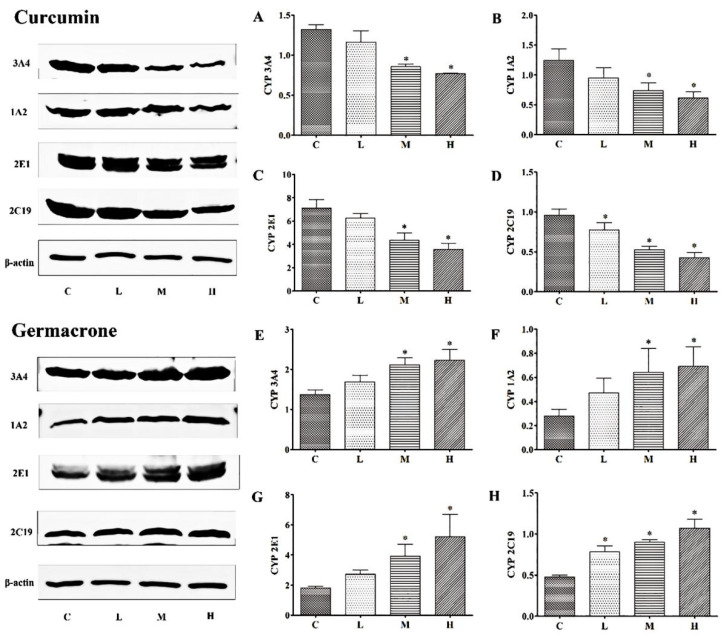
The expression levels of CYP3A4 (**A**), 1A2 (**B**), 2E1 (**C**) and 2C19 (**D**) were decreased in curcumin groups and CYP3A4 (**E**), 1A2 (**F**), 2E1 (**G**) and 2C19 (**H**) were increased in germacrone groups in vivo (mean ± SD, *n* = 6, * *p* < 0.05 vs. control rats).

**Table 1 molecules-27-04482-t001:** Metabolic parameters of curcumin and germacrone incubated in RLMs and HLMs (mean ± SD, *n* = 3).

Matrix	Compd	V_max_ (nM/min/mg of Protein)	k_m_ (μM)	CL_int_ (mL/min/(mg of Protein))	*r* ^2^
RLM_S_	Curcumin	15.63 ± 1.50	48.45 ± 8.27	0.3226	0.9978
HLMs	Curcumin	4.03 ± 0.54	16.55 ± 4.30	0.2435	0.9934
RLM_S_	Germacrone	13.70 ± 1.76	49.58 ± 11.12	0.2763	0.9985
HLM_S_	Germacrone	14.78 ± 1.51	44.81 ± 5.47	0.3298	0.9989

## Data Availability

The data presented in this study are available on request from the corresponding author.

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
