# Peer review of "Kinetic Characteristics of Curcumin and Germacrone in Rat and Human Liver Microsomes: Involvement of CYP Enzymes"

_molecules, 2022, doi:10.3390/molecules27144482_

Round 1
Reviewer 1 Report
The authors have written the manuscript entitled Discovery of the bioactive compounds of curcumin and germacrone selective in RLMs and HLMs in a very sequential and scientific way. This manuscript is well-designed and well-described and covered all necessary parameters. Although, there are many flaws and areas which should be improved before publishing it.
1. The tittle is not appealing for readers Remove abbreviations from the title.
2. Extensive revision is required in abstract, as the present sentences sounds noisy during reading. There are grammatical errors and need to minimize the sentences in length. Also please delete the first sentence from the abstract. Overall the abstract is not informative enough and need to show the actual picture of the work.
3. Line 38; Zingiberaceae family plant such as Curcuma aromatica Salisb., C. longa L.. [ Ref] this line describing curcumin should be supported by an appropriate reference as it look incomplete without coding any reference, this for guidance like. https://doi.org/10.3390/molecules27082468, DOI: 10.3390/molecules26237168; DOI: 10.1007/s00044-019-02497-0;
4. Add latest relevant references.
5. Rephrase the sentence (line 40) to understand it better, “The big health costs for NSCLC drug in clinical now”.
6. Line 53, 55, 57; provide space after references.
7. Line 77-78 should be rephrased for better understanding by the readers.
8. The introduction could have a figure with the structures of the most relevant bioactive compounds cited in it (RLMs and HLMs), together with the structure of curcumin and germacrone, to provide the readers an insight of the field and to situate the obtained compounds.
9. The authors should explain the procedure for extraction of plant materials/ should add a scheme for the extraction and isolation of pure compounds.
10. In the discussion section line 403 to 419 was without coding any appropriate literature/previous finding. The author only commented of their results. It is a scientific study; the author should support their findings with the appropriate literature.
11. The conclusion of the study have different format from the rest of the paper. It should be according to the journal format.
12. The article should be checked by a native speaker for improving and removing spelling and grammatical mistakes.
13. In my opinion, the article has more information for future perspective; it should be revised for the mentioned points before publishing.
Author Response
(1) To the first peer reviewer:
Discovery of the bioactive compounds of curcumin and germacrone selective in RLMs and HLMs. The authors have written the manuscript entitled “Discovery of the bioactive compounds of curcumin and germacrone selective in RLMs and HLMs” in a very sequential and scientific way. This manuscript is well-designed and well-described and covered all necessary parameters.
Thanks a lot for your approval!
Q1. The tittle is not appealing for readers Remove abbreviations from the title.
A: About this question, after thinking over and referencing specialized knowledge, we decide to change it as ‘‘Discovery of the bioactive compounds of curcumin and germacrone selective in rat and human liver microsomes’’.
Q2. Extensive revision is required in abstract, as the present sentences sounds noisy during reading. There are grammatical errors and need to minimize the sentences in length. Also please delete the first sentence from the abstract. Overall the abstract is not informative enough and need to show the actual picture of the work.
A: Thank you for pointing out the issue, the first sentence of the abstract has been deleted, and others sentences or letters have been changed to show the actual picture of this work.
Q3. Line 38; Zingiberaceae family plant such as Curcuma aromatica Salisb., C. longa L.. [Ref] this line describing curcumin should be supported by an appropriate reference as it look incomplete without coding any reference, this for guidance like. https://doi.org/10.3390/molecules27082468, DOI: 10.3390/molecules26237168; DOI: 10.1007/s00044-019-02497-0.
A: Yes, add an appropriate reference can be more completed. The added appropriate reference is [2] Oglah, M.K.; Mustafa, Y.F. Curcumin analogs: synthesis and biological activities. Med. Chem. Res. 2019, doi: 10.1007/s00044-019-02497-0], and the order of rear references are changed accordingly.
Q4. Add latest relevant references.
A: The latest relevant references were updated, such as [10, 11, 15, 16, 38, 39, 40].
Q5. Rephrase the sentence (line 40) to understand it better, “The big health costs for NSCLC drug in clinical now”.
A: “The big health costs for NSCLC drug in clinical” has been changed to “the NSCLC cancer need an expensive treatment in clinical” in the reviewed manuscript.
Q6. Line 53, 55, 57; provide space after references.
A: In lines 53, 55 and 57, the necessary space have been added.
Q7. Line 77-78 (meaning that CYP is an important prerequisite for the biological transformation of large of substances, and these enzymes have obviously genetic diversity) should be rephrased for better understanding by the readers.
A: Sorry the lines have been changed in order, but the sentence has been rephrased as ‘‘meaning that CYP enzyme is the main prerequisite for the biological transformation, and difference enzymes are controlled by genetic diversity’’.
Q8. The introduction could have a figure with the structures of the most relevant bioactive compounds cited in it (RLMs and HLMs), together with the structure of curcumin and germacrone, to provide the readers an insight of the field and to situate the obtained compounds.
A: Firstly, references 23 and 24 are the relativity for the bioactive compounds cited in RLMs and HLMs; secondly, the structures of curcumin and germacrone (Figure 1, which was Figure S1 in original manuscript) is added in the end of introduction. And others Figures order are changed accordingly in the reviewed manuscript.
Q9. The authors should explain the procedure for extraction of plant materials/ should add a scheme for the extraction and isolation of pure compounds.
A: Both the extraction of plant materials and the purified procure for the two compounds have added in the reviewed manuscript. Meanwhile, adding “2.2 Extraction, Isolation and identification” title in the reviewed manuscript, and the follow title orders were changed.
Q10. In the discussion section line 403 to 419 was without coding any appropriate literature/previous finding. The author only commented of their results. It is a scientific study; the author should support their findings with the appropriate literature.
A: The references of 38, 39 and 40 have been added in relative section to support the findings in the reviewed manuscript.
Q11. The conclusion of the study have different format from the rest of the paper. It should be according to the journal format.
A: OK, the format in conclusion has been changed to fit the journal format.
Q12. The article should be checked by a native speaker for improving and removing spelling and grammatical mistakes.
A: In the original manuscript, there were some spelling and grammatical errors. In the modified manuscript, we checked again and again, and we hope it goes well in content and format.
Q13. In my opinion, the article has more information for future perspective; it should be revised for the mentioned points before publishing.
A: We hope the revised manuscript can fit the request of Molecules, thank you!
Reviewer 2 Report
Manuscript molecules- 1782645 investigated the bioactivity of curcumin and germacrone on human microsomes and rat microsome. The following items should be either added or clarified:
1. Please reformat the content, make sure the font is consistent throughout the manuscript.
2. There should be a section in materials and methods that describe the extraction and isolation method of curcumin and germacrone, or cite proper reference in section 2.1 materials, line 92.
3. Please state the age of the SD rats.
4. Please state what the IS working solution it is in section 2.9.
5. Please move the stability validation method in section 3.1 to section 2.3. And please also explain the reason that fresh stock (0 day) was not included in the stability test.
6. Please provide the formulation of accuracy and precision calculation.
7. Please add blank RLM and HLM chromatograms in Figure.1 to prove no endogenous or exogenous interference.
8. Lines 270-271, please correct “The accuracy of curcumin and germacrone in HLMs were 88.4296.47%” to “The accuracy of curcumin and germacrone in HLMs were 88.42-96.47%”
9. Please add error bar to every data in Figure 2 and state the conc. of both compounds in the title of Figure 2.
10. Please clearly state the legends of Figure 6. It is very confusing what “C,L,M,H” stands for in very linear chart.
Author Response
(2) To the second peer reviewer:
Q1. Please reformat the content, make sure the font is consistent throughout the manuscript.
A: The font and space have been formated to fit the request throughout the manuscript.
Q2. There should be a section in materials and methods that describe the extraction and isolation method of curcumin and germacrone, or cite proper reference in section 2.1 materials, line 92.
A: The relative content has been added in the “materials and methods” section to describe the extraction and isolation method of curcumin and germacrone.
Q3. Please state the age of the SD rats.
A: 8 weeks SD rats has been added.
Q4. Please state what the IS working solution it is in section 2.10.
A: The IS working solution has been added in section 2.10.
Q5. Please move the stability validation method in section 3.1 to section 2.4. And please also explain the reason that fresh stock (0 day) was not included in the stability test.
A: â‘ The section 3.1 has been moved to section 2.4. â‘¡ The stability for fresh stock solution (0 day) was investigated in our study, but it was not expression in this article according to unnecessary, 1 month is long to show the liquid stabolyty. In other references, the stability can stand for 3 month (Etoxazole is Metabolized Enantioselectively in Liver Microsomes of Rat and Human in Vitro. Environmental science & technology, 2016: 9682. doi: 10.1021/acs.est.6b02676).
Q6. Please provide the formulation of accuracy and precision calculation.
A: The formulation of accuracy and precision calculation have been provided in supplementary material in the original supplement material. Accuracy% = X/T × 100% (X: measured value, T: average), Precision calculation: RSD% = SD/mean × 100%
Q7. Please add blank RLMs and HLMs chromatograms in Figure.1 to prove no endogenous or exogenous interference.
A: The blank RLMs and HLMs chromatograms were added to Figure 2 (original Figure 1 changed to Figure 2).
Q8. Lines 270-271, please correct “The accuracy of curcumin and germacrone in HLMs were 88.4296.47%” to “The accuracy of curcumin and germacrone in HLMs were 88.42-96.47%”.
A: This question is a typo. We modified to 88.42‒96.47% in the modified manuscript.
Q9. Please add error bar to every data in Figure 2 and state the conc. of both compounds in the title of Figure 2.
A: â‘ In Figure 3 (Figure 2 changed to Figure 3), the error bar to every data has been exist, some big and some too little to show low level. â‘¡ The concentrations of both compounds in the title of Figure 3 have been added.
Q10. Please clearly state the legends of Figure 6. It is very confusing what “C,L,M,H” stands for in very linear chart.
A: In the Figure 7 (original manuscript Figure 6 changed to reviewed Figure 7), the “C, L, M, H” have been legend clearly.

Reviewer 3 Report
Please refer to the observations suggestions available as notes in the enclosed file.

Author Response
(3) To the third peer reviewer:
Q1. implies several parameters including absorption, metabolism (phases I and II), distribution and excretion. Pharmacokinetics may be explored in whole living organism, since you worked only on microsomes you could study only the aspects related to the phase I metabolism of these compounds. Thus the term pharmacokinetic is not appropriate.
A: This research was focused on kinetics, so “pharmacokinetics” has been changed to “kinetics”. Thank you so much for the professionalism!
Q2. Who performed the botanical identification? One representative sample of the studied materials has been retained for any further reference?
A: This question has been answered in the reviewed manuscript.
Q3. Check the spelling. It should be as Exactive.
A: Thank you for pointing out this issue. Yes, it should be as Q Exactive (Thermo Fisher Scientific Q Exactive Mass spectrometer). It is was a typo mistake.
Q4. Why did you not performed the study "in vivo" by directly using the rats?
A: Compared with in vivo, in vitro studies require fewer animals and avoid interference from endogenous substances, meanwhile the role of drugs and targets were monitored clearly. The metabolic pathways and metabolites of drugs may be different in vitro and in vivo, and polymorphisms of CYP enzymes are very important to the individual differences in drug reactions. Besides, the blood pharmacokinetics, urodynamics with curcumin or germacrone in rats were reported (1. Pharmacokinetics and pharmacodynamics of three oral formulations of curcumin in rats. doi: 10.1007/s10928-020-09675-3. 2. Ameliorative role of curcumin on copper oxide nanoparticles﹎ediated renal toxicity in rats: an investigation of molecular mechanisms. doi: 10.1002/jbt.22593. 3. Pharmacokinetics of a single dose of turmeric curcuminoids depends on formulation: results of a human crossover study. doi: https://doi.org/10.1093/jn/nxab087.)
Q5. Check the meaning of this sentence (their RSD < 5%, and the stock solution curcumin or germacrone was stabled in 1 month)?
A: This question has been checked, and we sure that the stock solution curcumin or germacrone was stabled in 1 month.
Q6. Add spacing?
A: It has been added space in the necessary positions round the manuscript.

Round 2
Reviewer 1 Report
The manuscript has been improved by the authors as per suggestions. I think the title is still not appealing for readers and needs improvement.The manuscript needs final proper check for grammar and spellings mistakes before acceptance.
Author Response
Dear peer reviewer,
Thank you so much for the professional advices. Here are the respond, please check them, and if there are any questions, do not hesitate to contact us.
Q1. I think the title is still not appealing for readers and needs improvement.
A1. After thought about it, we change the topic as: Kinetic characteristics of curcumin and germacrone in rat and human liver microsomes: involvement of CYP enzymes.
Q2. The manuscript needs final proper check for grammar and spellings mistakes before acceptance.
A2. About the spell and grammar, we check again and again in this modify manuscript, and we hope it can fit the journal of Molecules.
Best wishes,
Jie Feng, Ph.D., Prof. of School of Pharmaceutical Science, Guangxi Medical University
July 3, 2022
